# Method of Directly Writing MPA on Photosensitive Surface of Detector Based on FIB

**DOI:** 10.3390/s24123769

**Published:** 2024-06-10

**Authors:** Anran Nie, Zhenwei Qiu, Xiaobing Sun, Jun Zhu, Jin Hong

**Affiliations:** 1Key Laboratory of General Optical Calibration and Characterization Technology, Hefei Institutes of Physical Science, Chinese Academy of Sciences, Hefei 230031, China; nieanran@mail.ustc.edu.cn (A.N.); xbsun@aiofm.ac.cn (X.S.); hongjin@aiofm.ac.cn (J.H.); 2University of Science and Technology of China, Hefei 230026, China; 3DFH Satellite Co., Ltd., Beijing 100094, China

**Keywords:** division of focal plane (DoFP) polarization detector, micro-polarizer array (MPA), crosstalk, focused ion beams (FIB)

## Abstract

The division of focal plane (DoFP) polarization detector has great potential for the development of aerospace polarimeters, but the existing commercial DoFP polarization detector cannot satisfy all the missions due to the diversity of satellite payloads. Here, we propose a method of directly writing a micro-polarizer array (MPA) on the detector surface based on focused ion beams (FIB) and fabricating a push-broom scanning DoFP polarization detector. The feasibility and low crosstalk of the solution were proved through testing, and the reasons for the low extinction ratio caused by oxidation were explained through characterization and numerical calculations. This scheme is not only applicable to DoFP polarization detectors but also provides ideas for the integration of other metasurface structures and detectors.

## 1. Introduction

Polarization is one of the fundamental properties of light waves. When a light wave interacts with a medium, the change in the polarization state reflects the surface or internal characteristics of the medium. Some creatures in nature can see the polarization of light, such as birds and mantis shrimps [1,2], which has become a powerful hunting weapon for them. Although the human eye is polarization-blind, the polarization characteristic of electromagnetic waves has been recognized by man-made polarizing devices and used in fields of medicine [3], image dehazing [4], navigation [5], and remote sensing [6], et al.

In the application of aerospace remote sensing and deep aerospace exploration, a part of satellite platforms carried polarization detection payloads [7,8,9,10,11,12]. The polarimeters of these payloads are mainly divided into the division of time (DoT) [7,8], the division of amplitude (DoAP) [9,10], and the division of focal plane (DoFP) [11,12] polarization detection. Among them, DoT and DoAP polarimeters are the most mature and widely used. However, the volume and weight of these types of polarimeters are relatively large, which is not conducive to lightweight and commercial satellites. DoFP polarization detection is a technology that integrates a micro-polarizer array (MPA) [13], which is based on sub-wavelength metal wire grids [14], on the surface of the image sensor. The polarimeter using this technology only needs a front-end single-channel optical system and a single-chip polarization detector, which greatly reduces the complexity, volume, and weight of the optical system. It is an important development direction for the future aerospace polarimeter.

At present, only SONY Corporation can provide the commercial DoFP polarization image sensor [15], and it has the best performance among similar products. However, due to the working wavelength band, full well capacity, pixel size, dynamic range, and other factors, SONY’s products cannot satisfy all aerospace missions. Therefore, according to different aerospace missions, different DoFP polarization detectors need to be manufactured. The methods of manufacturing DoFP polarization detectors can be divided into two types: one-step integration and distributed integration. One-step integration introduces the MPA fabrication craft in the process of manufacturing the image sensor [15,16]. Although this solution can ensure the polarizing performance of the detector, other types of polarization detectors require the development of a new detector manufacturing process. The high cost is not suitable for research and small-batch device production. Distributed integration is to fabricate the required MPA on a certain substrate and then paste it on the detector [17,18]. Due to issues such as crosstalk caused by the thickness of the adhesive layer [19], alignment of detector pixels and MPA, and grating collapse caused by stress during the adhesive process, the polarization performance of the DoFP polarization detector has significantly declined. Therefore, it would be a better choice to directly fabricate MPA on the surface of the photosensitive surface of the detector, and some research groups have already tried this solution [20].

The difficulty in fabricating MPA directly on the detector’s photosensitive surface is the choice of processing method. Packaged detectors need to be protected from high temperatures, pressure, and corrosive chemicals to prevent performance degradation or short-circuiting. Commonly used methods for fabricating MPA, such as laser interference lithography and electron beam exposure, require development, etching, and glue removal processes. The chemical reagents and gases involved in these processes can cause irreversible damage to the detector, so they are not suitable for fabricating microstructures on the detector’s photosensitive surface. Focused ion beam (FIB) [21,22] directly writing is a micro–nanoprocessing technology that uses high-energy ions to bombard the material surface to vaporize and peel the material. Since this technology is a purely physical process, it does not require the assistance of chemical reagents, ensuring that the detector is not damaged during fabrication. In addition, FIB has three-dimensional processing capabilities and almost no limits on the characteristic parameters of the microstructure. Therefore, we proposed a method that is based on (FIB) directly writing MPA on the photosensitive surface of commercial image sensors. A push-broom scanning DoFP polarization detector for aerospace remote sensing was designed, fabricated, measured, and characterized. We analyzed and calculated the problems that occurred during processing.

## 2. Sensor Construction and Fabrication Method

### 2.1. Sensor Construction

The DoFP polarization detector consists of a commercial monochromatic CMOS image sensor (CIS) and a self-developed MPA. In principle, any CIS or area array detector can be used to fabricate DoFP polarization detectors. However, polarimeters usually have specific spectral configurations, so monochrome CIS is more suitable for corresponding research. Additionally, most CIS are equipped with a microlens array to improve quantum efficiency, but adding MPA above the microlens array introduces additional crosstalk or ghosting [15]. Finally, due to the limitation of sample size on the FIB processing platform (the equipment model is Nova 200 Nanolab, FEI, USA), the packaged CIS needs to be less than 20 mm × 20 mm. Therefore, we have chosen monochrome CIS (CMV4000-3E5M0PN [23], AMS, Austria) for this research. On the one hand, this detector has no microlens array and contains a removable window glass; on the other hand, the CMV4000 is a sensor that has been used in aerospace missions [24].

The schematic diagram of the push-broom scanning DoFP polarization CIS (DPC) is illustrated in Figure 1a. Owing to that, the polarizing performance of subwavelength aluminum gratings is optimal in the visible and near-infrared bands, which has been confirmed by numerous studies [14,25]. Therefore, aluminum gratings are also used in this work. A certain-thick Al film was deposited on the photosensitive surface of the CIS to fabricate the subwavelength metal wire grids with polarization directions of 0°, 60°, and 120°. Theoretically, when the duty cycle of subwavelength Al gratings is 0.5, the period is smaller, and the height is higher, the TM transmittance and extinction ratio (ER) will both be higher. However, the actual grating parameters are determined by the fabricating capabilities. According to fabricating capabilities, the period, line width, and height of the gratings are 200 nm, 100 nm, and 150 nm, respectively. Figure 1c shows the simulation results of the sub-wavelength Al grating with this parameter, which is calculated by a commercial software package (Lumerical FDTD Solutions 2020 R2.4). Through FDTD calculations, the average ER of the grating with this parameter can exceed 400. Figure 1b shows the cross-sectional view of the designed polarimetric detector. Ideally, each grating area would need to perfectly match a row of pixels, but this requires markers that locate the pixels to achieve this goal. Since commercial CIS does not have markers, an over-matching method is adopted, which uses a wider grating area to match one row of pixels. Considering the pixel size (5.5 μm) and process deviation, the width W_1_ of each grating area is designed to be 11 μm, which ensures at least one row of pixels is completely covered by grating. The distance W_2_ between adjacent grating areas is 11 μm, which is intended to reduce optical and electrical crosstalk. The length L of the grating area is 110 μm. In addition, relying on the high displacement accuracy of the FIB processing platform and the direct writing accuracy of FIB, the boundary of the grating area is parallel to the boundary of the detector to ensure that there is no obvious angle between the grating area and the pixels.

The MPA of the traditional snapshot DPC adopts a structure like the Bayer color filter [15,17,18]; that is, a superpixel is composed of polarizers with polarization directions of 0°, 45°, 90°, and 135°. The four adjacent pixels correspond to different spatial positions. When the polarization in the target scene changes spatially, the four pixels measure different spatial polarization information. Merging the polarization information of these four pixels will greatly reduce the spatial resolution, and the interpolation method will lead to errors in the polarization information [26]. Therefore, the MPA with a traditional structure is not suitable for aerospace polarization remote sensing that pursues both spatial resolution and polarization accuracy. However, the detection target in aerospace remote sensing changes slowly and can even be considered stationary in a short time. According to the flight direction y, the pixels with different polarization angles of the push-broom scanning DPC pass over the detection target in sequence. All the polarization information of the target can be obtained without reducing the spatial resolution or introducing additional polarization errors. The degree of linear polarization (DoLP) and angle of polarization (AoP) need to be calculated [27].
(1)$$DoLP=\frac{\sqrt{Q^{2}+U^{2}}}{I^{2}}$$
(2)$$AoP=\frac{1}{2}\arctan(\frac{U}{Q})$$
(3)$$\left\{\begin{array}{c} I=(I_{0}+I_{60}+I_{120})/3\\ Q=(2I_{0}-I_{60}-I_{120})/3\\ U=(I_{60}-I_{120})/\sqrt{3} \end{array}\right.$$
where the symbols I, Q, and U refer to the Stokes vector elements and the $I_{0,\;60,120}$ represent the light intensity of light waves passing through the 0°, 60°, and 120° grating areas, respectively.

### 2.2. Fabrication Method

Figure 2 illustrates the whole fabrication process of DPC. Remove the window glass on the CIS surface to expose the photosensitive surface of the detector (Figure 2b). Normally, CIS packaging is completed in an ultra-clean semiconductor workshop, so the cleanliness of the photosensitive surface is up to standard and no additional cleaning is required. A 150 nm thick Al film was deposited on the photosensitive surface of the CIS by electron beam evaporation (EBE), as shown in Figure 2c. The grain size and density of the Al film directly affect the surface morphology and final performance of the grating. By controlling the deposition rate of Al, that is, low-speed deposition, a high-quality Al layer can be obtained. Place the Al-coated CIS on the FIB operating platform. After alignment, write the grating structure shown in Figure 1a directly on the Al layer (Figure 2d). FIB technology does not require development, etching, or glue removal processes, so the final required micro–nanopattern structure can be obtained after direct writing. Re-cover the window glass on the fabricated CIS surface to avoid physical attack and dust pollution, as shown in Figure 2e,f.

## 3. Performance Measurement

### 3.1. Radiometric Calibration

The radiometric calibration correlates the measured radiance to the detector output, which is necessary for the DoFP polarization detector. The transmittance of the grating area can be calculated by comparing the radiometric calibration of the CIS before and after fabrication. The radiometric calibration system consists of an integrating sphere built in a dark room and a CIS to be measured [28]. Fix the gain (1.0) and integration time (2 ms) of the CIS, adjust the brightness of the light outlet of the integrating sphere, and then collect the signal of the pixels under illumination. Shut down the light and then collect the signal without illumination, which is the dark current response of the CIS. Subtract the non-illuminated signal from the illuminated signal to obtain the response of the CIS to illumination. Most of the pixels of the fabricated CIS are covered by Al, which cannot respond to incident light. Therefore, it is only necessary to compare the calibration data of the pixels below the grating area.

Figure 3 illustrates the radiometric calibration results of CIS before and after grating fabrication, where the abscissa is the radiance at the exit of the integrating sphere and the ordinate is the digital number (DN) of the detector response. The solid square, circle, and triangle marks in the figure represent the radiometric calibration data before fabricating the 0°, 60°, and 120° grating areas, respectively. These data almost overlap and are relatively small in the standard deviation, indicating that the consistency of the CIS pixel response is good. Therefore, these three sets of data can be used for linear fitting curves with the same parameters. The fitted curve is black, and the slope and intercept are 0.38 and 10.76, respectively. The hollow square, circle, and triangle marks represent the radiation calibration data after fabricating the 0°, 60°, and 120° grating areas, respectively. The average DN values of all pixels in each grating area are lower than the data before fabrication, which proves that the transmittance of the grating is less than 1. In addition, the standard deviation of the DN value increased significantly after fabrication, which proves that the grating is not fabricated uniformly. Figure 3 also shows the radiometric calibration curves and equations for each grating area. The ratio of the slope of the color curve to the black curve can be approximated as the transmittance of the grating to unpolarized white light. The transmittances of the 0°, 60°, and 120° grating areas are, respectively, 82.29%, 75.34%, and 81.87%.

### 3.2. Polarizing Performance 

Polarizing performance is characterized by the transmittance of linearly polarized light in different polarization directions. A general polarization test system consists of a light source, an integrating sphere, a rotatable linear polarizer, and a DoFP polarization detector to be tested [29]. In this paper, we not only used a traditional polarization test system but also built a linear polarization measurement system with a line-type light source to study the impact of crosstalk on polarization performance. The linear polarization test system with a line-type light source is shown in Figure 4, which includes a halogen lamp (HL), a monochromator (MM), a collimator, a rotatable linear polarizer (RLP), a cylindrical mirror (CM), a slit, a microscope objective (MO), and a detector to be tested. The white light emitted by the HL passes through the MM, collimator, and RLP in sequence to form monochromatic, collimated, linearly polarized light. The light then passes through the CM and forms a narrow, bright line at the focal point. The slit is set at the focus position of the CM to eliminate stray light in the system. The width of the narrow, bright lines focused by CM is on the order of millimeters, which is much larger than the pixel size. Therefore, an MO needs to be set behind the slit to reduce the width of the bright line to 5.5 μm, which just covers a row of pixels. By translating the detector, the narrow, bright lines are imaged in different grating areas, and the linear polarizing plate is rotated to test the polarizing performance of the DPC.

By gradually changing the angle of the RLP from 0° to 360°, the DN of detectors was recorded for every 5°. The transmittance is the ratio of the DN value of DPC to the DN value of the original CIS under the same test conditions (eliminating the influence of dark current). Figure 5a illustrates the polarization transmittance of the wide-band integrating sphere light source, which is measured by a general polarization test system. Figure 5b–d shows the polarization transmittance results at wavelengths of 490 nm, 670 nm, and 865 nm, respectively, which were measured by our proposed test system. These three wave bands are the polarization spectrum channels of the atmospheric polarimeter, which are mainly used to study aerosols [6]. As the angle of linearly polarized light changes, the transmittance of all curves exhibits obvious sinusoidal properties. Compared with the curves of the 0° grating and the 60° grating, the extreme values of the 120° grating in Figure 5 are relatively high, proving that the polarizing performance of the 120° grating area is weak, and this will be analyzed in the next chapter. The curve in Figure 5a reflects the collective response of DPC to polarized light of various wavelengths and incident angles, while the curves in Figure 5b–d reflect the response of DPC to polarized light of a single wavelength and less than 15° incident angles. Combining the polarization transmittance curves, DPC exhibits better polarization performance for long-wave polarized light with small incident angles.

### 3.3. Crosstalk Measurement

Furthermore, we also measured the crosstalk, as shown in Figure 6a. Since the narrow light spot only illuminates one row of pixels, the response produced by the unilluminated pixels is crosstalk [30]. Taking the 60° grating area as an example, the upper part is composed of two rows of dark pixels and a 0° grating area, and the downer part is composed of two rows of dark pixels and a 120° grating area. It can be seen from Figure 6b that the dark pixels closest to the 60° grating area have high DN values, and the responses of these pixels are sinusoidal. The response of dark pixels next to the 60° grating area drops significantly but still has two peaks. The response of the 0° and 120° grating areas is extremely low. The upper surface of the dark pixels is covered with Al, which can completely block the incident light. Therefore, the response of the dark pixels comes from the diffracted light of the 60° grating area and the diffusion and drift of carriers in the photodiodes. If dark pixels are not set but three grating areas are closely connected, serious crosstalk will affect the polarization performance of each grating area. However, it can be corrected by this polarization measurement system. The average crosstalk of the 0° grating area and the 120° grating area are 0.54% and 0.59%, respectively, which are calculated by the ratio of the DN of non-illuminated pixels to the DN of illuminated pixels.

## 4. Characterization Results and Discussion

The measurement results in the previous chapter proved the feasibility of fabricating DoFP polarization detectors by directly writing MPA on the CIS surface based on FIB technology. However, the polarizing performance of the device is poor, and the 120° grating area has high transmittance and low ER. These phenomena need to be analyzed through characterization.

The ER of DPC depends on the structural parameters of the subwavelength metal grating and the crosstalk of MPA. The result in Figure 6b shows that the push-broom scanning DPC has extremely low crosstalk, which proves that the low ER of DPC is caused by the grating parameters. In fact, the theoretically calculated ER of the Al grating designed in this paper (period, line width, and height are 200 nm, 100 nm, and 150 nm, respectively) can reach 400. The difference between the actual measured value and the theoretical value proves that there were some problems in the grating fabrication. In addition, the polarizing performance of the 120° grating area is very weak, which also reflects fabrication problems. The reasons for these problems need to be found through characterization.

Figure 7a is a low-resolution scanning electron microscope (SEM) image of MPA. There are obvious differences between the image of the 120° grating and other grating areas. This is because the 120° grating area is the last grating area to be etched, and a long time working for FIB will produce thermal attenuation, resulting in a reduction in etching depth. Although the etching depth of the 120° grating area is shallow, it shows relatively high transmittance in both radiation calibration and polarization transmittance measurement. The only reason that can explain this phenomenon is that the Al between the grating gaps is oxidized to form aluminum oxide, which increases the transmittance and weakens the polarizing performance. In general, the thickness of the Al oxide film is several nanometers and dense, which can effectively protect the internal Al from further oxidation. Gallium (Ga) was used as the ion source in this fabrication. During the FIB writing process, a small amount of Ga will remain on the surface of the etching location. Ga and Al both belong to Group IIIA, so residual Ga can penetrate the interior of Al. Compared with the natural oxidation of pure Al, the oxide film generated by the Ga-Al alloy is loose, which cannot protect the internal Al from oxidation [31]. Therefore, as shown in Figure 7b, the thickness of the aluminum oxide is greatly increased, which explains the phenomenon of weak polarization and high transmittance in the 120° grating area. Figure 7c,d is are the middle-resolution SEM images of the 0° grating area and the 60° grating area, respectively. Since high-resolution SEM will cause slight melting of the Al grating, no high-resolution characterization was performed on the 0° and 60° grating areas. Comparing the two images, some black shadows appear in the 60° grating area, while there are almost no shadows in the 0° grating area. This shows that the etching depth is shallow in some parts of the 60° grating area, causing the transmittance and ER of this grating area to be inferior to those of the 0° grating area. The fabrication of the 0° grating area is relatively ideal, but the ER is still not high. In fact, the oxidation involving the Ga element occurs simultaneously in all grating areas.

As shown in Figure 8a, Ga elements will also remain on the side walls of the grating, causing the wire grid to produce an aluminum oxide film with a thickness of d, which will cause the grating duty cycle to decrease, the transmittance to increase, and the ER to decrease. Since there is almost no crosstalk between grating areas, the FDTD model of gratings is used for calculations. The real and imaginary parts of the refractive index of Al [32] at a wavelength of 865 nm are 1.6675 and 6.7544, respectively. Calculate the polarization transmittance of gratings with different alumina thicknesses, from d = 5 nm to d = 50 nm, respectively. Figure 8b shows the calculated results for various oxide layer thicknesses. As the thickness of the oxide layer increases, the minimum transmittance of the grating gradually increases, resulting in a decrease in ER. When d = 40 nm, the calculated results are closest to the measured data, which can indicate that the thickness of the oxide layer is about 40 nm. The DPC measurement results and SEM images fully prove the feasibility of using FIB to directly write MPA on the detector surface. However, ideal polarizing performance was not obtained due to the oxidation of the device. This can be solved by changing the ion source or adding a passivation layer on the surface of the MPA.

Currently, crosstalk has little impact on the ER due to the poor polarizing performance of the device. In the future, when the oxidation problem is solved, ultra-high ER can be obtained by optimizing grating parameters. At that time, crosstalk will be the most important factor affecting the ER. The internal structure of the CIS is one of the sources of crosstalk and an important basis for affecting the arrangement of MPA. Therefore, we combined mechanical cutting and FIB technology to prepare thin-section samples of CIS and characterized the CIS using SEM and X-ray fluorescence (XRF). Figure 9a is the SEM image of the internal structure of CMV4000, and Figure 9b–d are the XRF images of different elements of the sample. Combining the structural diagram and elemental analysis, CMV4000 is a front-side illuminated (FSI) CIS consisting of a silicon dioxide passivation layer containing circuit structures and a Si substrate layer. Circuit structures in the passivation layer can reflect incident photons into other pixels, causing increased crosstalk. For ER with a high extinction ratio, you can choose to appropriately increase the number of rows of dark pixels to reduce crosstalk. In addition, the thickness of the passivation layer of FISCIS is approximately 2–3 microns, which is also the main source of crosstalk caused by diffraction. So, backside-illuminated (BSI) CIS with a thinner passivation layer is more suitable for DPC fabrication.

ER is the most important indicator for evaluating polarization devices. The measured ER value of the DoFP polarization detector is determined by the intrinsic ER of the grating and the crosstalk. The intrinsic ER of gratings is affected by material and structural parameters. Crosstalk is affected by the distance between the MPA and the photodiode, as well as the internal microstructure of the detector. When the reciprocal of crosstalk is higher than the intrinsic extinction ratio of the grating, the measured ER is approximately equal to the intrinsic ER, and vice versa. The method proposed in this paper aims to reduce the distance between the MPA and the photodiode and successfully achieve low crosstalk. Although the polarization performance of the device is unsatisfactory, through characterization analysis and numerical calculations, the reasons for weak polarizing performance in the 120° grating area, low extinction ratio, and partial sources of crosstalk were identified, respectively. According to these issues, corresponding improvement methods have been proposed. Compared with previous work, this paper pays more attention to the crosstalk of DoFP polarization detectors and successfully fabricated a low-crosstalk DPC. In addition, the high geometric accuracy of the FIB equipment facilitates the registration of MPA and detector pixels, which eliminates the need for assistance from other devices during registration.

## 5. Conclusions

In this paper, we have proposed a method to directly write MPA on the CIS photosensitive surface based on FIB and manufacture a push-broom scanning DPC with different polarization angles (0°, 60°, and 120°). The polarizing performance and crosstalk of the detector were measured by using an improved linear polarization test system with a line-type light source. The results show that our DPC crosstalk is extremely low and quantifiable, and the system can measure and calibrate crosstalk for scanning the DoFP polarization detector. Although the polarizing performance of the device is poor, the feasibility of the proposed method is proven. Through the characterization analysis of MPA and CIS, the reason for the unsatisfactory ER due to grating oxidation was found. Protection methods such as replacing the FIB ion source or creating a passivation layer on the MPA surface to prevent grating oxidation are proposed. The results of this paper not only provide some references for the structural design and fabrication method of polarization detectors but also inspire ideas for the integration of other metasurface structures and detectors.

## Figures and Tables

**Figure 1 sensors-24-03769-f001:**
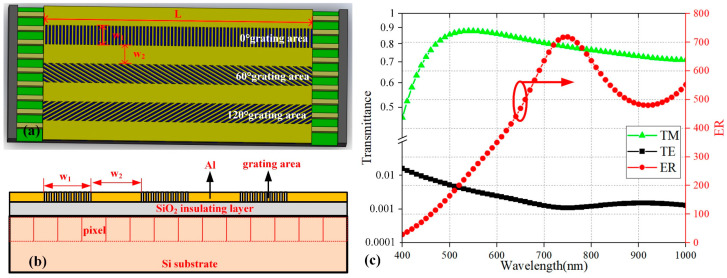
(**a**) Schematic diagram and (**b**) cross-sectional view of the designed push-broom scanning DPC, (**c**) Transmittance and ER curve with the grating period 200 nm, line width 100 nm, and height 150 nm. The value of ER is the ratio of TM wave transmittance to TE wave transmittance.

**Figure 2 sensors-24-03769-f002:**
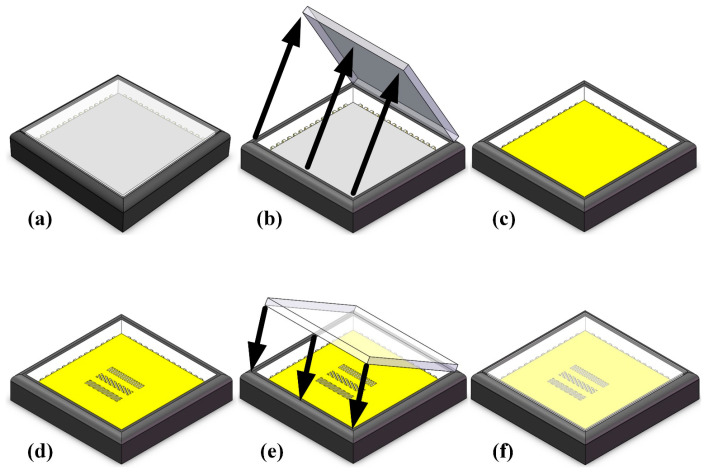
Schematic diagram of the fabrication process of the DoFP polarization detector. (**a**) Original CIS; (**b**) remove window glass; (**c**) deposit Al on the photosensitive surface of CIS; (**d**) FIB direct writing on the Al; (**e**) recover window glass; (**f**) final DPC schematic.

**Figure 3 sensors-24-03769-f003:**
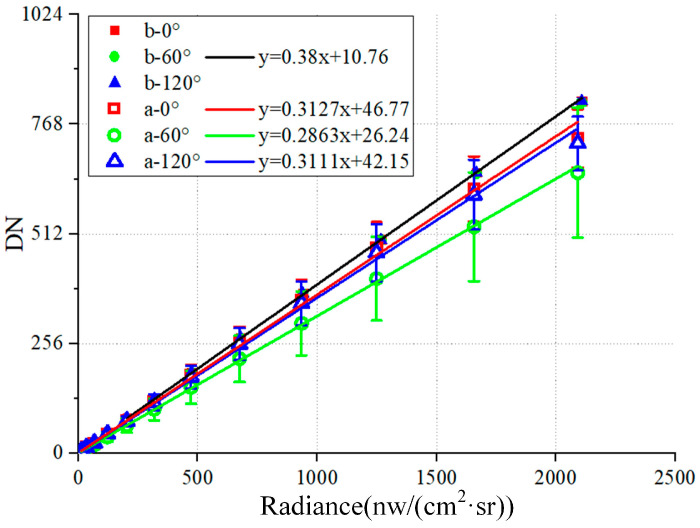
The radiometric calibration test results and fitting curves. The solid square, circle, and triangle marks represent the radiometric calibration data before fabricating the 0°, 60°, and 120° grating areas, respectively. The hollow square, circle, and triangle marks represent the radiation calibration data after fabricating the 0°, 60°, and 120° grating areas, respectively.

**Figure 4 sensors-24-03769-f004:**
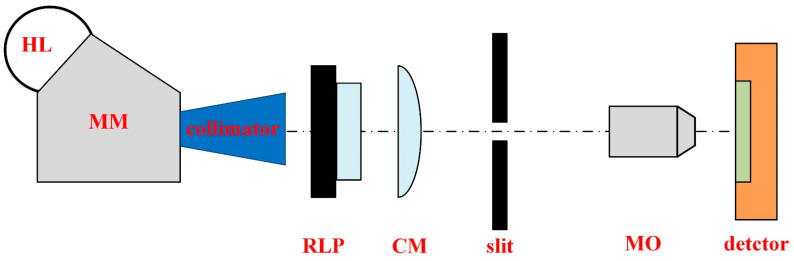
The linear polarization test system uses a line-type light source. HL, halogen lamp; MM, monochromator; RLP, rotatable linear polarizer; CM, cylindrical mirror; MO, microscope objective.

**Figure 5 sensors-24-03769-f005:**
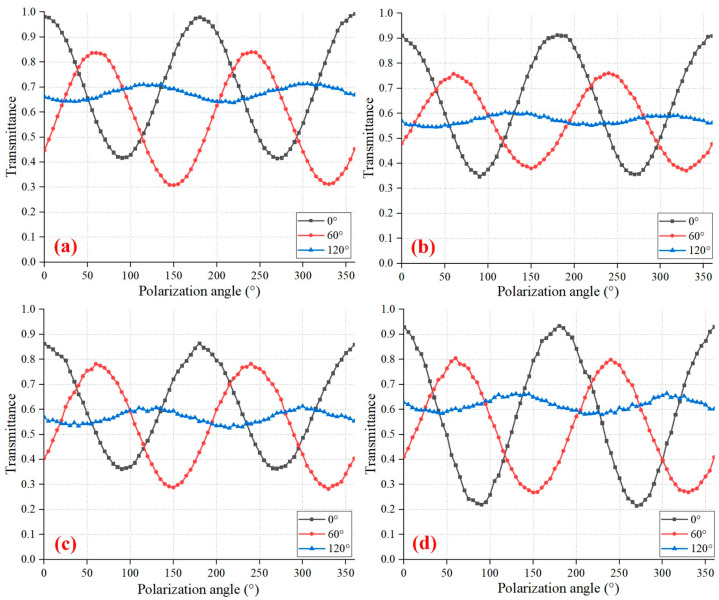
Polarization transmittance at different wavelengths. (**a**) white light (350 nm−1000 nm) polarized light, measured by a general polarization test system; (**b**) 490 nm; (**c**) 670 nm; (**d**) 865 nm, measured by a linear polarization test system with a line-type light source.

**Figure 6 sensors-24-03769-f006:**
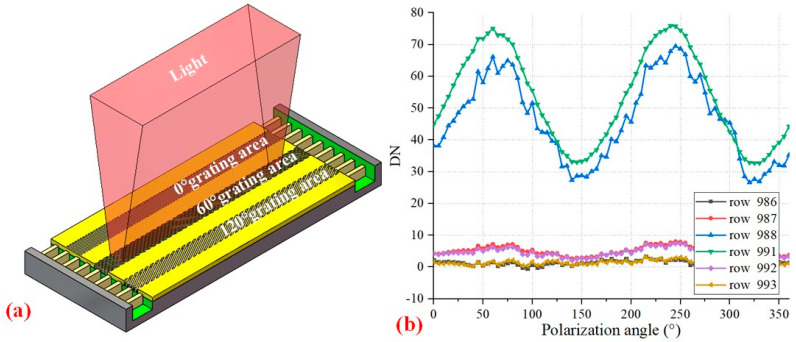
Crosstalk measurement principle and results. (**a**) Crosstalk measurement principle; (**b**) Crosstalk measurement result. From top to bottom, the pixel types (coordinates) are 0° grating area (row 986), dark pixels (rows 987 and 988), 60° grating area (rows 989 and 990), dark pixels (rows 991 and 992), and 120° pixel area (row 993). Irradiate row 989 to obtain the DN values from rows 986 to 988, and illuminate row 990 to obtain the DN values from rows 991 to 993.

**Figure 7 sensors-24-03769-f007:**
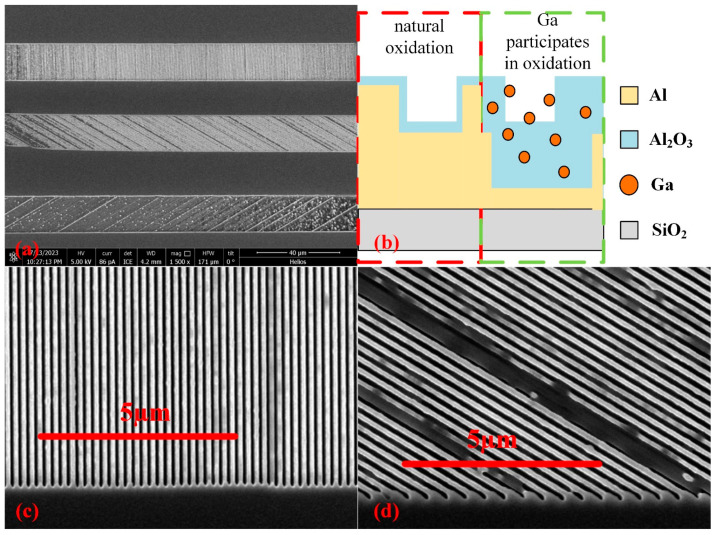
(**a**) Low-resolution SEM images of the MPA; (**b**) schematic diagram of the 120° grating area oxidation structure; (**c**) medium-resolution SEM images of the 0° grating area; (**d**) medium-resolution SEM images of the 60° grating area.

**Figure 8 sensors-24-03769-f008:**
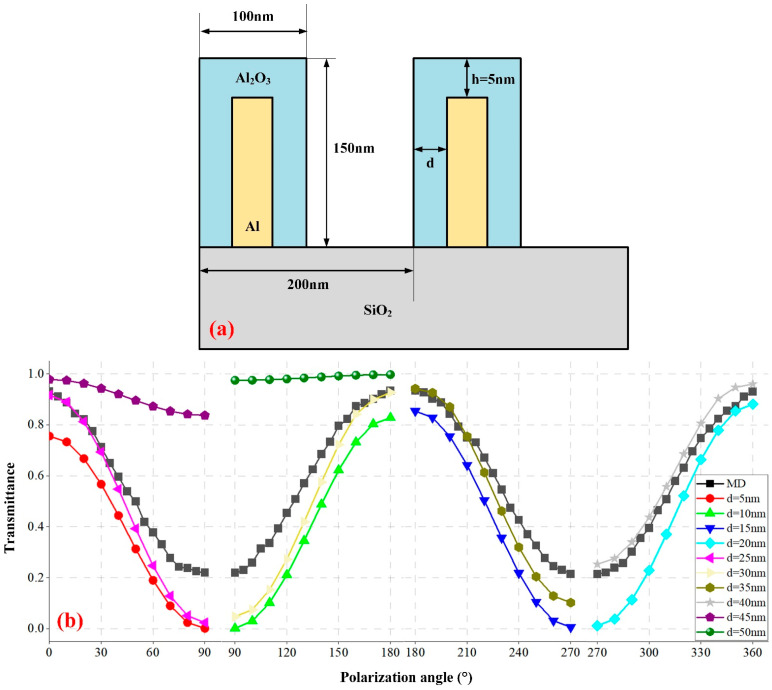
(**a**) FDTD model of grating with an oxide layer. h = 5 nm means that the top of the grating is naturally oxidized without the participation of the Ga element. (**b**) Polarized transmittance of gratings with different oxide layer thicknesses. The black curve is the measured dataset (MD) of the 0° grating area @865 nm, and the other curves are simulation data.

**Figure 9 sensors-24-03769-f009:**
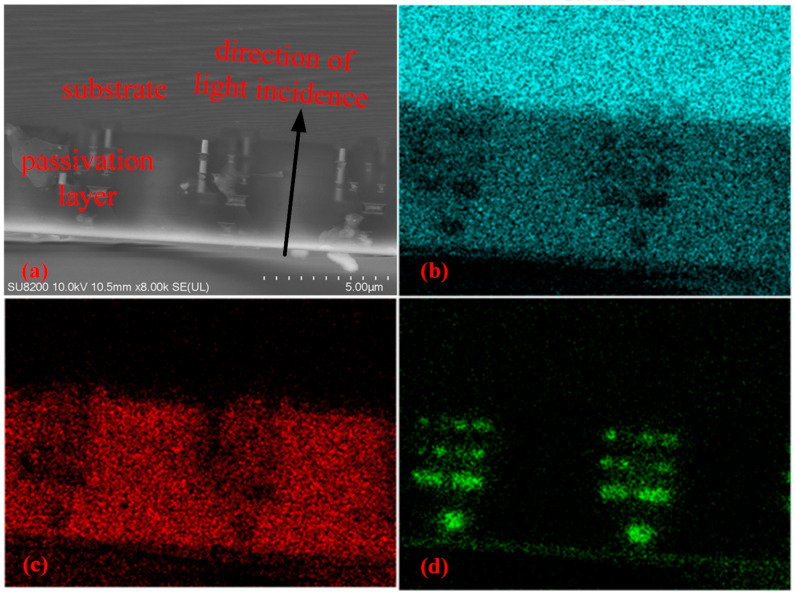
(**a**) SEM image of the internal structure of CMV4000. (**b**) XRF mapping of Si Kα1, (**c**) XRF mapping of O Kα1, (**d**) XRF mapping of Al Kα1. It can be judged that the passivation layer is composed of an Al circuit structure and silicon dioxide, and the substrate is made of Si.

## Data Availability

Data are available upon request from the authors.

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
