# Peer review of "Method of Directly Writing MPA on Photosensitive Surface of Detector Based on FIB"

_sensors, 2024, doi:10.3390/s24123769_

Round 1

Reviewer 1 Report

Comments and Suggestions for Authors

The authors of the article "Method of directly writing MPA on photosensitive surface of 2 detector Based on FIB" present a method of manufacturing micro-polarizer array structures on the surface of a commercial sensor using FIB technology. Although the presented method was used for aerospace remote sensing and deep aerospace exploration applications, it can also be used for other devices that have structures with dimensions comparable to the wavelength. The presented work is interesting but it cannot be used to manufacture devices on a large scale.

My comments and questions for the authors are the following:

1. When an optical device is develop, usually, a preliminary analysis is performed in which the optimal configuration is studied through numerical simulations, in most of the cases. The authors mention on page 3 that FDTD calculations were used without going into details. I believe that it is necessary to detail the method and to introduce in the paper few details such us: software name (commercially or not) ; input and output parameters; how the extinction ratio was calculated.

2. It is not clear how the aluminum grating values were decided. The FDTD simulations only refer to the extinction ratio.

3. I did not find data about the length of the grating.

4. Can authors estimate the manufacturing duration of one grating area ?

5. At point 2.2 Fabrication Method, the authors describe the steps used to fabricate the grating. Being a manufacturing process without etching, the FIB technology must be used for the total removal of the Al film up to the surface of the sensor. Did the authors check if the entire Al film was removed? How did they proceed to avoid destroying the surface of the sensor but also to completely remove the Al film?

6. What are the slit dimensions  ?

Author Response

Dear reviewer

We highly appreciate the detailed valuable comments from the referees on our manuscript. The suggestions are quite helpful, and we have incorporated them in the revised version of manuscript. And we hope that the reviewers and the editor will be satisfied with our responses to these comments and the revisions for the original manuscript.

In the section of comments and suggestions for authors, you mentioned that FIB is not suitable for large-scale manufacture devices. We strongly agree with your opinion. Our proposed method is suitable for small-scale manufacturing as well as laboratory studies. Therefore, in the section of introduction of the manuscript, we mentioned aerospace remote sensing. The detectors used in aerospace remote sensing are often specially customized and do not have general commercial value. Therefore, our manufacturing solution is still suitable.

Regarding the other questions, we have made targeted revisions to the manuscript. Please see the attachment for specific reply.

Finally, thank you again for your review of the manuscript. I wish you good health and luck.

Reviewer 2 Report

Comments and Suggestions for Authors

Article deals with fabrication and studies of micro-polarizer array (MPA) on detector surface. Dimensions of Al grating 100/100 nm are too narrow for simple optical lithography and authors use FIB for its fabrication.  Quality of such MPA is not very impressive, authors mention in the conclusion “polarizing performance of the device is poor”.  Much better quality of line resolution can be obtained with electron lithography even with low resolution. Another issue is the price of fabrication using FIB compared to electron lithography, the latter seems to be cheaper.  Authors present a list of 32 references, but did not present comparison of performance of their polarizer and different competitors.  It is not clear what is the goal of the research, what accuracy of polarization measurements is required. 

Author Response

Dear reviewer

We highly appreciate the detailed valuable comments from the referees on our manuscript. The suggestions are quite helpful, and we have incorporated them in the revised version of manuscript. And we hope that the reviewers and the editor will be satisfied with our responses to these comments and the revisions for the original manuscript.

I have summarized your comments and suggestions to the author into 3 questions.

  1. Why not use electron beam lithography (EBL)?
  2. Why not compare with other similar products?
  3. It is not clear what is the goal of the research, what accuracy of polarization measurements is required. 

Please see the attachment for specific reply.

Finally, thank you again for your review of the manuscript. I wish you good health and luck.
